# Prior Anti-Angiogenic TKI-Based Treatment as Potential Predisposing Factor to Nivolumab-Mediated Recurrent Thyroid Disorder Adverse Events in mRCC Patients: A Case Series

**DOI:** 10.3390/biomedicines11112974

**Published:** 2023-11-04

**Authors:** Luigi Liguori, Angelo Luciano, Giovanna Polcaro, Alessandro Ottaiano, Marco Cascella, Francesco Perri, Stefano Pepe, Francesco Sabbatino

**Affiliations:** 1Oncology Unit, Department of Clinical Medicine and Surgery, University of Naples “Federico II”, 80131 Naples, Italy; luigi.liguori2@unina.it (L.L.); angelo.luciano2@unina.it (A.L.); 2Oncology Unit, Department of Medicine, Surgery and Dentistry, University of Salerno, 84081 Baronissi, Italy; gpolcaro@unisa.it (G.P.); spepe@unisa.it (S.P.); 3SSD Innovative Therapies for Abdominal Metastases, Abdominal Oncology, INT IRCCS Foundation “G. Pascale”, 80131 Naples, Italy; a.ottaiano@istitutotumori.na.it; 4Unit of Anesthesiology, Intensive Care Medicine, and Pain, Department of Medicine, Surgery and Dentistry Medicine, University of Salerno, 84081 Baronissi, Italy; mcascella@unisa.it; 5Medical and Experimental Head and Neck Oncology Unit, INT IRCCS Foundation “G. Pascale”, 80131 Naples, Italy; f.perri@istitutotumori.na.it

**Keywords:** anti-angiogenic, ICI, mRCC, nivolumab, PD-1, recurrent, TDAEs, TKIs

## Abstract

Immune checkpoint inhibitors (ICIs) targeting programmed cell death 1 (PD-1) or its ligand 1 (PD-L1) have revolutionized the management of many types of solid tumors, including metastatic renal cell carcinoma (mRCC). Both sequential and combinatorial therapeutic strategies utilizing anti-PD-1 monoclonal antibodies (mAbs) and anti-angiogenic tyrosine kinase inhibitors (TKIs) have demonstrated to improve the survival of patients with mRCC as compared to standard therapies. On the other hand, both ICIs and TKIs are well known to potentially cause thyroid disorder adverse events (TDAEs). However, in the context of sequential therapeutic strategy, it is not clear whether prior anti-angiogenic TKI may increase the risk and/or the severity of ICI-related TDAEs. In this work, by describing and analyzing a case series of mRCC patients treated sequentially with prior TKIs and then with ICIs, we investigated the role of prior anti-angiogenic TKI-based treatment as a potential predisposing factor to anti-PD-1-mediated recurrent TDAEs, as well as its potential impact on the clinical characteristics of nivolumab-mediated recurrent TDAEs. Fifty mRCC patients were included in the analysis. TKI-mediated TDAEs were reported in ten out of fifty patients. TKI-mediated TDAEs were characterized by hypothyroidism in all ten patients. Specifically, 40%, 40% and 20% of patients presented grade 1, 2 and 3 hypothyroidisms, respectively. Following tumor progression and during anti-PD-1 nivolumab treatment, five out of ten patients developed anti-PD-1 nivolumab-mediated recurrent TDAEs. Anti-PD-1 nivolumab-mediated recurrent TDAEs were characterized by an early transient phase of thyrotoxicosis and a late phase of hypothyroidism in all five patients. The TDAEs were grade 1 and 2 in four and one patients, respectively. Prior anti-angiogenic TKI did not modify the clinical characteristics of nivolumab-mediated recurrent TDAEs. However, all five patients required an increased dosage of levothyroxine replacement therapy. In conclusion, our work suggests that prior anti-angiogenic TKI-based treatment significantly increases the risk of ICI-mediated recurrent TDAEs in patients with mRCC without modifying their clinical characteristics. The most relevant effect for these patients is the need to increase the dosage of lifelong levothyroxine replacement therapy.

## 1. Introduction

Immune checkpoint inhibitor (ICI)-based immunotherapy has dramatically changed the management of many types of solid tumors, including metastatic renal cell carcinoma (mRCC) [1,2,3,4,5]. This novel immunotherapeutic approach improves the tumor response and survival outcomes of patients with mRCC as compared to standard therapies [3,4,5]. Anti-programmed cell death 1 (PD-1) monoclonal antibody (mAb) nivolumab has been the first ICI approved for the treatment of mRCC, demonstrating to improve the overall survival (OS) and safety profile of mRCC patients refractory to prior anti-angiogenic tyrosine kinase inhibitor (TKI)-based treatment as compared to everolimus [3]. More recently, the combinations of anti-PD-1 or anti-programmed death ligand 1 (PD-L1) mAbs with anti-cytotoxic T-lymphocyte-associated antigen 4 (CTLA-4) mAb or anti-angiogenic TKIs have received approval for the treatment of naïve mRCC [5,6,7,8,9]. Nevertheless, not all patients achieve a long-term benefit from ICI-based immunotherapy, and a non-negligible portion of treated patients develop immune-related adverse events (irAEs) [3,5,6,7,8,9,10,11]. The latter are autoimmune conditions, which can potentially affect any organ in the body, including endocrine glands, mimicking “de novo” autoimmune diseases [12,13]. irAEs can affect any organ and differ in terms of the time of onset, severity and clinical course [12,13]. Thyroid disorder adverse events (TDAEs) have emerged as some of the most common endocrine irAEs, with the prevalence ranging from 3.15% to 30.00% in patients treated with ICIs. TDAEs are more commonly reported in patients treated with anti-PD-1 monotherapy or in combination with anti-CTLA-4 as compared to anti-CTLA-4 monotherapy [14,15,16,17].

In addition, TDAEs are also reported in cancer patients, including mRCC patients, treated with anti-angiogenic TKIs, such as axitinib, cabozantinib, pazopanib, sorafenib, sunitinib and tivozanib. In mRCC patients, TKI-related TDAEs are reported in about 14.00–31.00% of patients depending on the drug used, diagnostic criteria and pre-treatment thyroid function. On the other hand, in mRCC patients treated with combinatorial strategies of ICIs and TKIs, an increased incidence of TDAEs (about 25–47% of patients) is reported as compared to TKI-related TDAE incidence [6,7,8,9].

In both cases, ICI- and TKI-related TDAEs are characterized by subclinical or overt primary hypothyroidism preceded by a transient phase of thyrotoxicosis. The latter is frequently undiagnosed due to an early onset, short duration and lack of symptoms [16,18,19,20,21,22]. In contrast, Graves’-disease-like presentations have only been reported as anecdotic evidence [23,24,25].

The pathophysiological mechanisms underlying the development of TDAEs are not completely clear. Some lines of evidence suggest that the cytotoxic effect of activated T cells against thyrocytes could explain ICI-related TDAEs [16,22,26]. On the other hand, TKI-related TDAEs are likely caused by the vascular damage of the thyroid gland induced by anti-angiogenic activity of TKIs [18,27,28].

The increased incidence of TDAEs in the combinatorial strategies of ICIs and TKIs prompted us to investigate whether, in the context of sequential strategy, prior anti-angiogenic TKIs increase the risk and/or severity of ICI-related TDAEs. Here, starting from the description of a case series, we investigated the role of prior anti-angiogenic TKI-based treatments as a potential predisposing factor to anti-PD-1 nivolumab-mediated recurrent TDAEs in patients with mRCC. In addition, by describing the clinical characteristics of nivolumab-mediated recurrent TDAEs, we also investigated the potential impact of prior anti-angiogenic TKI-based treatments on the clinical characteristics of nivolumab-mediated recurrent TDAEs. Lastly, the main aim of this study was to increase the state of the literature on nivolumab-mediated recurrent TDAEs in mRCC patients in order to improve their clinical management.

## 2. Case Presentation

We performed a retrospective analysis of mRCC patients who were sequentially treated with prior TKIs and then with ICIs at San Giovanni di Dio e Ruggi D’Aragona University Hospital between May 2016 and May 2018. The study was performed without interfering with clinical practice. The selection of patients to be included in the study was performed based on (i) signed informed consent for clinical–pathological data acquisition; (ii) age > 18 years; (iii) treatment with anti-PD-1 nivolumab as a second-line treatment following TKI-based treatment; (iv) informed consent for clinical data collection. Clinical data—including age, sex, Eastern Cooperative Oncology Group (ECOG) performance status (PS), smoking status, comorbidities, tumor histology, presence/absence of asymptomatic brain metastases, serum values of TSH, fT3, fT4, TgAbs, TPOAbs and TRAbs before first nivolumab administration—were retrieved. Radiographic imaging was performed every 8 weeks. The response rate was determined according to the Response Evaluation Criteria in Solid Tumours version 1.1 (RECIST v1.1) [29]. The irAEs were reported according to the Common Terminology Criteria for Adverse Events (CTCAE) v 4.0 [30]. Patients who died from COVID-19 were excluded from the study. The study was approved by the local ethics committee (prot./SCCE n.85275), in accordance with the Declaration of Helsinki and its amendments. All data were collected using Microsoft Excel. In order to investigate the role of prior anti-angiogenic TKI-induced TDAE as a potential predisposing factor to anti-PD-1 nivolumab-mediated recurrent TDAEs, we first selected patients who developed TDAE during treatment with TKIs. Second, among these patients, we selected those who developed nivolumab-mediated recurrent TDAE and described their clinical characteristics.

Fifty patients were included in the analysis (the main baseline clinical characteristics of the study populations are summarized in Table 1). Of those, ten patients (20.0%) developed TDAEs during treatment with TKIs. Following tumor progression, these patients received second-line nivolumab treatment, and five out of ten (50.0%) developed recurrent TDAEs. The clinical course of these five patients and their clinical–pathological characteristics are summarized in Table 2.

Among patients who did not previously develop TKI-mediated TDAEs, only 2 out of 40 (5.0%) developed nivolumab-mediated TDAEs. All patients received nivolumab (3 mg/kg every two weeks) by intravenous infusion as second-line therapy according to the national and European Society for Medical Oncology (ESMO) guidelines for the treatment of mRCC. Sunitinib and pazopanib were administered as previous TKIs in 80% and 20% of patients, respectively. The main clinical characteristics of nivolumab-mediated recurrent TDAEs are summarized in Table 3.

Nivolumab-mediated recurrent TDAEs were characterized by an early transient phase of thyrotoxicosis and a late phase of hypothyroidism in all five patients. The median time of the onset of thyrotoxicosis after the first nivolumab administration was 21 days (range: 14–469 days), while the median duration of transient thyrotoxicosis was 15 days (range: 10–21 days). Hypothyroidism occurred in a median time of 20 days (range: 14–49 days) from the onset of thyrotoxicosis. The median time for the normalization of serum values of thyrotropin (TSH), free triiodothyronine (fT3) and free thyroxine (fT4) was 25 days (range: 15–42 days). Grade 1 and grade 2 TDAEs occurred in four and one patients, respectively. In addition, both thyrotoxicosis and hypothyroidism were reported as asymptomatic and symptomatic in four and one patients, respectively. After the thyrotoxicosis phase, all patients required an increased dosage of levothyroxine replacement therapy to restore normal serum values of TSH, fT3 and fT4. The levels of anti-thyroglobulin antibodies (TgAbs), anti-thyroid peroxidase antibodies (TPOAbs) and anti-thyrotropin receptor antibodies (TRAbs) were negative throughout the whole clinical course in all five patients. Complete responses (CRs), partial responses (PRs) and progressive diseases (PDs) were reported in one (20%), two (40%) and two (40%) of five treated patients, respectively. Three out of five patients were still alive at the last follow-up in June 2023. As described in Table 3, these patients globally presented similar clinical characteristics as compared to patients who developed nivolumab-induced TDAE with no previous TKI-mediated TDAE. In the following subheadings, we describe in more detail the clinical course of these five patients.

### 2.1. Patient 1

The first patient was a 53-year-old man with no relevant comorbidities, first diagnosed in June 2016 with sarcomatoid RCC (sRCC) in an advanced stage of disease due to the presence of multiple liver, lung and bone metastases. He was classified as intermediate risk according to the International Metastatic Renal Cell Carcinoma Database Consortium (IMDC) risk score. The patient received pazopanib (800 mg/day) in combination with denosumab (120 mg every 28 days) as first-line treatment for advanced disease. In August 2016, the whole-body CT scan showed disease progression, with an increase in the number and size of lung lesions, as well as an increase in the size of the primary renal lesion. After one month of treatment, the patient developed grade 2 TDAE. As a result, he started levothyroxine replacement therapy (50 μg/day), which resulted in normalization of serum values of TSH, fT3 and fT4. The lack of response to pazopanib prompted second-line treatment with nivolumab (3 mg/kg every 14 days) in combination with denosumab (120 mg every 28 days). The patient achieved a prolonged partial response from nivolumab with no irAEs. Following 469 days from the start of nivolumab administration, the patient developed asymptomatic thyrotoxicosis for 10 days. Subsequently, a subclinical hypothyroidism occurred after 14 days from the onset of thyrotoxicosis. By increasing levothyroxine replacement therapy (from 50 μg/day to 75 μg/day), following another 25 days, the serum values of TSH, fT3 and fT4 were normalized. Nivolumab was administrated throughout the whole period of ICI-related TDAE. During further treatment with nivolumab in the next three months, the patient developed severe Guillain–Barrè-like syndrome (grade 3) and hepatitis (grade 4), leading to discontinuation of nivolumab administration and prompt administration of immunosuppressive therapy (intravenous immunoglobulin (IVIG) and mycophenolate mofetil, respectively). Following 7 years of follow-up, the patient is still alive with no disease progression and no further irAEs. However, he still needs to continue levothyroxine replacement therapy (75 μg/day).

### 2.2. Patient 2

A 60-year-old man with type II diabetes and no further relevant comorbidity was diagnosed in May 2017 with clear cell RCC (ccRCC) in an advanced stage due to the presence of multiple lung and infradiaphragmatic and supradiaphragmatic lymph node metastases. He was classified as intermediate risk according to the IMDC risk score. The patient received first-line treatment with sunitinib (50 mg/day for 4 weeks, followed by 2 weeks without treatment). Following five weeks of treatment, the patient developed grade 1 TDAE. As a result, he started levothyroxine replacement therapy (50 μg/day), which resulted in the normalization of serum values of TSH, fT3 and fT4. Following further nine months with relative disease control, the whole-body CT scan showed disease progression, with an increase in the number and size of lung and lymph node metastases. Due to disease progression, the patient started nivolumab (3 mg/kg every 14 days) as second-line treatment. Following 49 days from the first administration of nivolumab, the patient developed asymptomatic thyrotoxicosis for 21 days. Subsequently, a subclinical hypothyroidism occurred 49 days after the onset of thyrotoxicosis. By increasing levothyroxine replacement therapy (from 50 μg/day to 75 μg/day), following 21 days, the serum values of TSH, fT3 and fT4 were normalized. Nivolumab was administrated throughout the whole period of the TDAE. The patient achieved a prolonged partial response from nivolumab with no other irAEs until he developed grade 4 bullous pemphigoid, leading to discontinuation of nivolumab administration and administration of high-dose steroid. Following six years of follow-up, the patient is still alive with no disease progression and no further irAEs. He still needs to continue levothyroxine replacement therapy (75 μg/day).

### 2.3. Patient 3

A 51-year-old man with no relevant comorbidity was first diagnosed in January 2017 with ccRCC in an advanced stage of disease due to the presence of multiple liver, spleen, lung and infradiaphragmatic lymph node metastases. He was classified as intermediate risk according to the IMDC risk score. The patient received first-line treatment with sunitinib (50 mg/day for 4 weeks, followed by 2 weeks without treatment), achieving a durable partial response lasting 15 months. Following two months from the start of sunitinib administration, the patient developed grade 3 TDAE. As a result, he started levothyroxine replacement therapy (50 μg/day), which resulted in normalization of serum values of TSH, fT3 and fT4. In May 2017, a whole-body CT scan showed disease progression, with an increase in the size and number of liver, lung and infradiaphragmatic lymph node metastases. Then, he started nivolumab (3 mg/kg every 14 days) as second-line treatment. Following 15 days from the start of nivolumab administration, the patient developed symptomatic thyrotoxicosis characterized by an increase in the heart rate and blood pressure for 15 days. Following further 19 days from the onset of thyrotoxicosis, overt hypothyroidism occurred. It was characterized by a decrease in the heart rate and blood pressure, as well as by weakness and drowsiness. Nivolumab administration was stopped. By gradually increasing levothyroxine replacement therapy (from 50 μg/day to 125 μg/day), after further 42 days, the patient did not present any hypothyroidism-related symptoms, and serum values of TSH, fT3 and fT4 were normalized. As a result, the patient re-started the administration of nivolumab. However, following three months from the start of nivolumab administration, a whole-body CT scan showed disease progression, with an increase in the size of liver and lung metastases and the occurrence of novel multiple brain metastases. Given the serious decline in his clinical conditions, the patient was transferred to another hospital in order to continue the best supporting care and died in the next 20 days.

### 2.4. Patient 4

A 72-year-old man with hypertension, prostatic hypertrophy and no further relevant comorbidity was first diagnosed in July 2016 with ccRCC in an advanced stage due to the presence of multiple liver and lung metastases. He was classified as intermediate risk according to the IMDC risk score. The patient received first-line treatment with sunitinib (50 mg/day for 4 weeks, followed by 2 weeks without treatment), achieving a durable partial response lasting 16 months. Following two months from the start of sunitinib, the patient developed grade 2 TDAE. Then, he started levothyroxine replacement therapy (50 μg/day), which resulted in normalization of serum values of TSH, fT3 and fT4. In October 2017, a whole-body CT scan showed disease progression, with an increase in the size and number of lung metastases. Then, he started second-line treatment with nivolumab (3 mg/kg every 14 days). Following 14 days from the start of nivolumab administration, the patient developed asymptomatic thyrotoxicosis for 14 days. Subsequently, a subclinical hypothyroidism occurred 20 days after the onset of thyrotoxicosis. By increasing levothyroxine replacement therapy (from 50 μg/day to 75 μg/day), following 30 days, the serum values of TSH, fT3 and fT4 were normalized. Nivolumab was administrated throughout the whole period of the TDAE. In December 2017, a whole-body CT scan showed partial response of all sites of disease. In March 2018, following nivolumab treatment, a whole-body CT scan showed a complete response of the disease. Following seven years of follow-up, the patient is still alive with no disease progression; he continues to receive nivolumab with no further irAEs. He still needs to continue levothyroxine replacement therapy (75 μg/day).

### 2.5. Patient 5

A 51-year-old man with bilateral glaucoma and no further relevant comorbidity was first diagnosed in April 2017 with ccRCC in an advanced stage due to the presence of multiple bone, liver, lung and infradiaphragmatic and supradiaphragmatic lymph node metastases. He was classified as poor risk according to the IMDC risk score. The patient received sunitinib (50 mg/day for 4 weeks, followed by 2 weeks without treatment) in combination with denosumab (120 mg every 28 days) as first-line treatment for advanced disease. Following 20 days of sunitinib treatment, the patient developed grade 1 TDAE. As a result, he started levothyroxine replacement therapy (25 μg/day), which resulted in the normalization of serum values of TSH, fT3 and fT4. In June 2017, a whole-body CT scan showed disease progression, with an increase in the number and size of all lesions. Then, he started second-line treatment with nivolumab (3 mg/kg every 14 days) and denosumab (70 mg/mL every 28 days). Following 21 days from the start of nivolumab administration, the patient developed asymptomatic thyrotoxicosis lasting 14 days. Subsequently, a subclinical hypothyroidism occurred 21 days after the onset of thyrotoxicosis. By increasing levothyroxine replacement therapy (from 25 μg/day to 50 μg/day), following further 15 days, the serum values of TSH, fT3 and fT4 were normalized. Nivolumab was administrated throughout the whole period of the TDAE. In September 2017, following three months from the start of nivolumab administration, the whole-body CT scan displayed disease progression, with the occurrence of new multiple liver and bone metastases. The clinical conditions of the patient worsened dramatically due to progression of the disease. The patient was transferred to another hospital in order to continue the best supporting care and died in the next 10 days.

## 3. Discussion

Both sequential and combinatorial therapeutic strategies utilizing ICIs and anti-angiogenic TKIs represent the cornerstone of the treatment of mRCC patients [6,7,8,9]. Nevertheless, more than half of the patients treated with ICIs do not achieve any clinical benefit, and roughly 10% of the patients develop severe endocrine irAEs, causing prolonged sequelae, such as the need for lifelong hormone replacement therapies [5,6,7,8,9]. TDAE is the most common endocrine irAE, leading to the need for hormone replacement therapy [14,15,16,17]. In the same manner, anti-angiogenic TKIs may also cause TDAEs, determining the need for hormone replacement therapy [27,28,31,32,33]. However, in the context of sequential therapeutic strategy, it is not clear whether prior anti-angiogenic TKI may increase the risk and/or severity of ICI-related TDAEs.

In the present work, we reported that 50% of patients with mRCC who received a prior anti-angiogenic TKI and developed TKI-mediated TDAE also developed nivolumab-mediated recurrent TDAE. In contrast, among patients who did not develop TKI-mediated TDAEs, only 5% of patients subsequently developed nivolumab-mediated TDAEs. In addition, by analyzing the clinical characteristics of those patients developing nivolumab-mediated recurrent TDAEs, a few differences were found. The phase of thyrotoxicosis was asymptomatic and symptomatic in four (80%) and one (20%) patients, respectively. In both cases, the diagnosis was difficult because of the absence of any or specific symptoms. As a result, our study strengthens the importance of an intensive evaluation of the serum levels of fT3, fT4 and TSH, regardless of thyroid symptoms, in patients treated with ICIs who received prior anti-angiogenic TKI. The median duration of thyrotoxicosis in our patients was 15 days, and it occurred in the first 50 days in most patients (80%). In contrast, one patient (20%) developed TDAE 469 days after the first administration of nivolumab. The median time to onset of hypothyroidism, as well as to normalization of the serum values of TSH, fT3 and fT4, was similar in all nivolumab-mediated recurrent TDAE patients. These characteristics of ICI-related TDAEs do not appear to be influenced by prior anti-angiogenic TKI [22,34,35]. Moreover, most patients (80%) did not present any hypothyroidism-related symptoms. Only one patient developed overt hypothyroidism, preceded by a symptomatic thyrotoxicosis phase. Notably, the same patient developed a symptomatic grade 3 TDAE from prior anti-angiogenic TKI. This association suggests that high-grade TDAE from prior anti-angiogenic TKI may predispose to high-grade nivolumab-mediated recurrent TDAE. In contrast, other evidence has shown no clear difference in the severity of ICI-induced TDAEs, regardless of prior TKI-induced TDAE [34,35]. Worthy of note, in our case, all patients developing a nivolumab-mediated recurrent TDAE required an increase in the dosage of levothyroxine replacement therapy in order to normalize their serum values of TSH, fT3 and fT4. The median increase in levothyroxine replacement therapy was 25 μg/day. In line with other evidence, all patients required an increase in the dosage of levothyroxine replacement therapy throughout their whole clinical course [22,36]. The serum levels of thyroid autoantibodies (TgAbs, TPOAbs and TRAbs) were negative in all five patients throughout the whole clinical course. At the time of last follow-up, 60% of the five patients developing nivolumab-mediated recurrent TDAE were still alive and continued to receive levothyroxine replacement therapy. Despite the small study population, our results suggest that patients who experienced prior TKI-related TDAE have a higher predisposition to develop nivolumab-mediated recurrent TDAEs, which require an increased dosage of levothyroxine replacement therapy.

All the main clinical characteristics of ICI-related TDAEs reported in our study are in line with data from the literature. Indeed, the median duration of thyrotoxicosis (15 days) confirms a rapid and more often asymptomatic nature of this phase [22,36,37,38]. Although far less frequent, the occurrence of late TDAEs has already been reported in the literature [22,39]. Whether prior anti-angiogenic TKI may influence the risk of late irAEs is not defined and should be further investigated. The onset of hypothyroidism, as well as the normalization of serum values of TSH, fT3 and fT4, is also in line with data reported in the literature [22,34,35,36]. Lastly, our results confirm that Graves’-like as well as Hashimoto-like diseases are rarely reported in patients treated with ICIs [23,24,25].

To the best of our knowledge, only Sbardella et al. demonstrated that prior TKIs may potentially predispose to ICI-related TDAEs in cancer patients [36]. Nevertheless, in their study, they enrolled patients with different types of cancer treated with different types of TKI, including both anti-angiogenic and anti-epidermal growth factor receptor (EGFR) TKIs. In contrast, our study only included mRCC patients pre-treated with anti-angiogenic TKIs. In our study, 80% and 20% of the patients, respectively, were first treated with sunitinib and pazopanib. The limited number of patients did not allow for drawing a clear conclusion on the specific effect of TKI. As a result, further studies will be needed to establish which TKI may potentially predispose more to ICI-mediated recurrent TDAEs.

According to the well-known role of angiogenesis in the homeostasis of the thyroid gland, we hypothesize that the anti-angiogenic activity of TKIs underlies their potential predisposing effect on ICI-mediated recurrent TDAEs. Angiogenesis is a process required for the growth of new vessels from pre-existing vasculature, which plays a major role in the promotion of tumor growth and metastasis spread [40,41,42]. This process is well regulated by several molecules, such as cytokines, growth factors, prostaglandins and proteolytic enzymes [40,43,44,45]. Among these molecules, vascular endothelial growth factors (VEGFs) and their receptors (VEGFRs) are some of the most widely investigated. The latter are expressed in many types of human tissue, including the thyroid gland, where they are also involved in the modulation of endocrine function by regulating the thyroid microvasculature [40,46]. VEGFRs are the main targets of anti-angiogenic TKIs currently utilized for the treatment of mRCC [47,48,49]. However, no clear evidence has demonstrated the definitive mechanism underlying the inhibition of the VEGF/VEGFR axis and thyroid dysfunction so far. In the last decade, different mechanisms have been proposed. For instance, both in vitro and in vivo studies have shown that the inhibition of VEGFRs leads to microvasculature regression of the thyroid gland, which in turn causes destructive thyroiditis and its dysfunction [49,50,51,52,53]. On the other hand, some studies have suggested further potential mechanisms involving the regulation of the iodine metabolism of the thyroid gland. For instance, sunitinib and sorafenib induce hypothyroidism by blocking the iodine uptake in the thyroid gland and by increasing type 3 deiodination, respectively [54,55]. In addition, an in vitro study by Braun et al. demonstrated that some TKIs (sunitinib, imatinib and dasatinib) may decrease the uptake of thyroid hormones by inhibiting monocarboxylate transporter 8 (MCT8) [56]. Nevertheless, how these alterations may increase the predisposition to ICI-mediated recurrent TDAEs should be further investigated. Lastly, in a recent study, Jeong et al. demonstrated that anti-angiogenic TKIs may induce PD-L1 upregulation through activation of the mTOR pathway in mRCCs [57]. We might speculate that the upregulation of PD-L1 on thyroid cells—as well as their altered immunogenicity induced by blocking of VEGFR-mediated pathway activation—may be the key to an explanation, given that prior anti-angiogenic TKI-based treatments increase the risk of ICI-mediated recurrent TDAE.

The main strength of this study is the presentation of real-world data. The latter are crucial in evaluating the efficacy/safety of novel treatments beyond the results of clinical trials, as well as generating novel hypotheses, such as the one in our study. However, the small sample size of the study population, the retrospective nature of the study and the lack of a clear mechanism are important limitations, which require further investigations.

## 4. Conclusions

In conclusion, our work suggests that prior anti-angiogenic TKI-based treatment increases the risk of ICI-mediated recurrent TDAE in patients with mRCC. In addition, prior anti-angiogenic TKI-based treatment does not seem to dramatically modify the clinical characteristics of ICI-related TDAEs. In our small study population, we reported that the most relevant effect is the need to increase the dosage of levothyroxine replacement therapy in order to normalize the serum values of TSH, fT3 and fT4. As a result, we suggest a closer observation of mRCC patients who received prior anti-angiogenic TKI in order to improve the management of ICI-mediated recurrent TDAEs. In this regard, the role of endocrinologists in the context of a multi-disciplinary team appears to be crucial. However, further studies enrolling larger study populations are needed to confirm whether prior anti-angiogenic TKI-based treatments increase the risk and severity of ICI-mediated recurrent TDAEs.

## Figures and Tables

**Table 1 biomedicines-11-02974-t001:** Baseline clinical–pathological characteristics of patients included in the study.

Median Age	69 Years (48–83 Years)
**Sex:**	
Male	42 (84.00%)
Female	8 (16.00%)
**ECOG PS:**	
0	20 (40.00%)
1	20 (40.00%)
2	2 (10.00%)
**Smoking Status:**	
Never smoker	3 (6.00%)
Previous smoker	33 (66.00%)
Current smoker	14 (28.00%)
**Comorbidities:**	
Hypertension	37 (74.00%)
Dyslipidemia	20 (40.00%)
Diabetes	15 (30.00%)
COPD	13 (26.00%)
HF	3 (6.00%)
Immune disorder	0 (0.00%)
**Previous Nephrectomy:**	
Yes	42 (84.00%)
No	8 (16.00%)
**IMDC Risk Score:**	
Favorable risk	12 (24.00%)
Intermediate risk	28 (56.00%)
Poor risk	10 (20.00%)
**Histology:**	
ccRCC	45 (90.00%)
sRCC	5 (10.00%)
Other	0 (0.00%)
**Asymptomatic Brain Metastases:**	
Yes	4 (8.00%)
No	46 (92.00%)
**Type of TKI:**	
Sunitinib	40 (80.00%)
Pazopanib	10 (20.00%)

Abbreviations: ccRCC: clear cell renal cell carcinoma, COPD: chronic obstructive pulmonary disease, ECOG: Eastern Cooperative Oncology Group, HF: heart failure, IMDC: International Metastatic Renal Cell Carcinoma Database Consortium, PS: performance status, sRCC: sarcomatoid renal cell carcinoma, TKI: tyrosine kinase inhibitor.

**Table 2 biomedicines-11-02974-t002:** Clinical–pathological characteristics of mRCC patients developing nivolumab-mediated recurrent TDAEs from previous TKIs.

	Patient 1	Patient 2	Patient 3	Patient 4	Patient 5
Age	53	60	51	72	51
Sex	Male	Male	Male	Male	Male
IMDC Risk Score	Intermediate	Intermediate	Intermediate	Intermediate	Poor
Histology	sRCC	ccRCC	ccRCC	ccRCC	ccRCC
Prior TKI	Pazopanib	Sunitinib	Sunitinib	Sunitinib	Sunitinib
Best Response to TKI	PD	PR	PD	PD	PD
TKI-Induced TDAE	Grade 2	Grade 1	Grade 3	Grade 2	Grade 1
Levothyroxine Replacement Therapy before First Nivolumab Administration	50 μg/day	50 μg/day	50 μg/day	50 μg/day	25 μg/day
Serum Values of TSH, fT3 and fT4 before First Nivolumab Administration	Normal	Normal	Normal	Normal	Normal
Levels of TgAbs, TPOAbs and TRAbs before First Nivolumab Administration	Negative	Negative	Negative	Negative	Negative

Abbreviations: ccRCC: clear cell renal cell carcinoma, fT3: free triiodothyronine, fT4: free thyroxine, IMDC: International Metastatic Renal Cell Carcinoma Database Consortium, PD: progressive disease, PR: partial response, sRCC: sarcomatoid renal cell carcinoma, TDAE: thyroid disorder adverse event, TgAbs: anti-thyroglobulin antibodies, TKI: tyrosine kinase inhibitor, TPOAbs: anti-thyroid peroxidase antibodies, TRAbs: anti-thyrotropin receptor antibodies, TSH: thyrotropin.

**Table 3 biomedicines-11-02974-t003:** The main clinical characteristics of nivolumab-mediated TDAEs in patients with mRCC.

**Patients with Previous TKI-Mediated TDAE**					
	**Patient 1**	**Patient 2**	**Patient 3**	**Patient 4**	**Patient 5**
Onset of Thyrotoxicosis (Days from First Nivolumab Administration)	469	49	15	14	21
Duration of Thyrotoxicosis (Days)	10	21	15	14	16
Onset of Hypothyroidism (Days from the Start of Thyrotoxicosis)	14	49	19	20	21
Time to Normalization of Serum Values of TSH, fT3 and fT4 (Days)	25	21	42	30	15
Clinical Course of Thyrotoxicosis	Asymptomatic	Asymptomatic	Symptomatic	Asymptomatic	Asymptomatic
Clinical Course of Hypothyroidism	Subclinical	Subclinical	Overt	Subclinical	Subclinical
Nivolumab-Mediated Recurrent TDAE	Grade 1	Grade 1	Grade 2	Grade 1	Grade 1
Levothyroxine Replacement Therapy	75 μg/day	75 μg/day	125 μg/day	75 μg/day	50 μg/day
Levels of TgAbs, TPOAbs and TRAbs	Negative	Negative	Negative	Negative	Negative
Best Response to Nivolumab	PR	PR	PD	CR	PD
**Patients with No Previous TKI-Mediated TDAE**					
		**Patient 7**		**Patient 8**	
Onset of Thyrotoxicosis (Days from First Nivolumab Administration)		42		22	
Duration of Thyrotoxicosis (Days)		20		15	
Onset of Hypothyroidism (Days from the Start of Thyrotoxicosis)		48		21	
Time to Normalization of Serum Values of TSH, fT3 and fT4 (Days)		30		20	
Clinical Course of Thyrotoxicosis		Asymptomatic		Asymptomatic	
Clinical Course of Hypothyroidism		Subclinical		Subclinical	
Nivolumab-Mediated TDAE		Grade 1		Grade 1	
Levothyroxine Replacement Therapy		50 μg/day		25 μg/day	
Levels of TgAbs, TPOAbs and TRAbs		Negative		Negative	
Best Response to Nivolumab		PR		SD	

Abbreviations: CR: complete response, fT3: free triiodothyronine, fT4: free thyroxine, PD: progressive disease, PR: partial response, SD: stable disease, TgAbs: anti-thyroglobulin antibodies, TPOAbs: anti-thyroid peroxidase antibodies, TRAbs: anti-thyrotropin receptor antibodies, TSH: thyrotropin, TDAEs: thyroid disorder adverse events.

## Data Availability

The raw data supporting the conclusions of this article will be made available by the authors without undue reservation.

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
