# Peer review of "Prior Anti-Angiogenic TKI-Based Treatment as Potential Predisposing Factor to Nivolumab-Mediated Recurrent Thyroid Disorder Adverse Events in mRCC Patients: A Case Series"

_biomedicines, 2023, doi:10.3390/biomedicines11112974_

Round 1

Reviewer 1 Report

Comments and Suggestions for Authors

I read this article with great interest. This is a case series about the occurrence opf TDAE in patietns affected by metastatic RCC treated before with TKi and then etih ICI (nivolumab). With the observations reported, the authors suggest that previous TKI TDAE predispose a second similar event with nivolumab, resulting in higher LT-4 dosage.
The manuscript is very well written. The work is well structured, clear in all its parts. The topics covered are consistent with each other. The table are clear and well structured, summarizing in a clear and precise way the rcases reported.

However, considering what has been said, I have only a few suggestions: 

- Line 97: written like this, it seems that the progression of the disease began and continued under therapy with nivolumab, when instead the patients switched to treatment with ICIs for the progression that had previously emerged with TKIs. This is why I ask the authors to modify this sentence

- Do the Authors have any data about the thyroid US features of the reported patients? Both before and during the development of TDAE

- Lines 270-271: From where this data come from? I did not find it the results section (5% vs 2.5%). Please revise accordingly.

-Patients n1 displayed some clinical aspects that made him different from the other ones: beyond the longer time span after what he developed the TDAE from nivolumab, he had also suffered from GBS and Hepatitis. Beyond thyroid antibodies, did he display other serological markers of autoimmunity? (ie ANA, ENA etc)

- The "Discussion" section appears quite disorganized, even if its content is well written and the comparison with the literature is complete. Therefore, I suggest Authors to modify its structure as follows: 1 synthesis of the background and the results of their study 2. comparison with the literature 3. speculation of the underlying pathogenetic mechanisms 4. Strength and limitations of the current study. 5 avoid repetitions (ie lines 272-273 and 309-310) and put future purpose in the "Conclusions" section (ie lines 297-298)

Author Response

Reviewer 1:

I read this article with great interest. This is a case series about the occurrence opf TDAE in patietns affected by metastatic RCC treated before with TKi and then etih ICI (nivolumab). With the observations reported, the authors suggest that previous TKI TDAE predispose a second similar event with nivolumab, resulting in higher LT-4 dosage.
The manuscript is very well written. The work is well structured, clear in all its parts. The topics covered are consistent with each other. The table are clear and well structured, summarizing in a clear and precise way the rcases reported.

However, considering what has been said, I have only a few suggestions: 

- Line 97: written like this, it seems that the progression of the disease began and continued under therapy with nivolumab, when instead the patients switched to treatment with ICIs for the progression that had previously emerged with TKIs. This is why I ask the authors to modify this sentence

We thank the Reviewer for really appreciating our manuscript. According to reviewer’s suggestion, we modified the sentence.

- Do the Authors have any data about the thyroid US features of the reported patients? Both before and during the development of TDAE

Unfortunately, no data about thyroid US features before TDAE are available. All five reported patients performed Thyroid US during the development of TDAE. However, non-specific thyroid US features have been detected in all five patients.  

- Lines 270-271: From where this data come from? I did not find it the results section (5% vs 2.5%). Please revise accordingly.

We accordingly revised the sentence in lines 270-271.

-Patients n1 displayed some clinical aspects that made him different from the other ones: beyond the longer time span after what he developed the TDAE from nivolumab, he had also suffered from GBS and Hepatitis. Beyond thyroid antibodies, did he display other serological markers of autoimmunity? (ie ANA, ENA etc)

A larger panel of serological markers of autoimmunity has been tested for the Patient 1 including: i) ANA, SMA, anti-LMK1 and anti- LC1 antibodies for autoimmune hepatitis; ii) anti-ganglioside antibodies for Guillan-Barrè syndrome and iii) anti-Jo-1, anti-ARS, anti-Mi-2, and anti-SRP antibodies for autoimmune myositis. No autoimmune alterations have been detected.   

- The "Discussion" section appears quite disorganized, even if its content is well written and the comparison with the literature is complete. Therefore, I suggest Authors to modify its structure as follows: 1 synthesis of the background and the results of their study 2. comparison with the literature 3. speculation of the underlying pathogenetic mechanisms 4. Strength and limitations of the current study. 5 avoid repetitions (ie lines 272-273 and 309-310) and put future purpose in the "Conclusions" section (ie lines 297-298)

According to reviewer’s suggestion, we modified the structure of discussion

Reviewer 2 Report

Comments and Suggestions for Authors

Dr. Liguori with colleagues presents an interesting paper that describes case series aiming to evaluate whether prior anti-angiogenic TKI-based treatment significantly increases the risk of ICI-mediated recurrent TDAEs in patients with mRCC. This paper touches truly important areas of medicine as well as has medium-to-high level of novelty. Thorugh the peer-review process some majors and minors must be fixed. Namely:

- Since the introduction provides really nice overview of immune checkpoint inhibitors, it lacks a clear statement of the research objectives and hypotheses that the paper aims to address. Bringing them to the table as the last paragraph of the intro would be good. 

- This paper describes case series, which is valuable since it is real-world human experience. However, it's important to clarify the selection criteria for the cases and why these specific cases were chosen. Also comparing the results to the negative/control group would be very beneficial for the paper. 

- Even for case series, the sample size of five patients with nivolumab-recurrent TDAEs is very small, which limits the generalizability of the findings. The Auhtors should definitely have more precaution when formulating their conclusions. 

- Another point, I want to raise is the fact that the paper briefly mentions a retrospective analysis but lacks detailed information on the methods used for data collection, inclusion/exclusion criteria, and statistical analysis. Providing a transparent and comprehensive desrciption of the methodology is essential for replication and evaluation by other researchers. Please re-consider it.

- Providing more basic characteristic of patients would bring more interest from the readers.

Minors:

Latin phrases like "in vitro" should go into italic font.

Data Availability Statement should be filled up since there is a set of analyzed data presented.

The paper has multiple, redundant minor grammatical errors and very awkward phrases. Please proofread carefully. Clear and concise language is essential for effective scientific communication.

Comments on the Quality of English Language

The paper has multiple, redundant minor grammatical errors.

Author Response

Reviewer 2:

Dr. Liguori with colleagues presents an interesting paper that describes case series aiming to evaluate whether prior anti-angiogenic TKI-based treatment significantly increases the risk of ICI-mediated recurrent TDAEs in patients with mRCC. This paper touches truly important areas of medicine as well as has medium-to-high level of novelty. Thorugh the peer-review process some majors and minors must be fixed. Namely:

- Since the introduction provides really nice overview of immune checkpoint inhibitors, it lacks a clear statement of the research objectives and hypotheses that the paper aims to address. Bringing them to the table as the last paragraph of the intro would be good. 

We thank the Reviewer for really appreciating our manuscript. According to Reviewer’s suggestion we added a clear sentence to state the paper aim. We put the sentence at the last paragraph of the intro.

- This paper describes case series, which is valuable since it is real-world human experience. However, it's important to clarify the selection criteria for the cases and why these specific cases were chosen. Also comparing the results to the negative/control group would be very beneficial for the paper. 

We better clarify the methodology of the study. We also better specify why have selected the patients for the description of their cases. In discussion we specify that the characteristics of nivolumab-recurrent TDAE are similar to those of patients with ICI-related TDAE without prior TKI-related TDAE.  

- Even for case series, the sample size of five patients with nivolumab-recurrent TDAEs is very small, which limits the generalizability of the findings. The Auhtors should definitely have more precaution when formulating their conclusions. 

According to Reviewer’s suggestion we highlighted the limitation of study population. In addition, we re-formulated our conclusions with more precaution.

- Another point, I want to raise is the fact that the paper briefly mentions a retrospective analysis but lacks detailed information on the methods used for data collection, inclusion/exclusion criteria, and statistical analysis. Providing a transparent and comprehensive desrciption of the methodology is essential for replication and evaluation by other researchers. Please re-consider it.

According to Rewiever’s suggestion we provide a more complete description of methodology.

- Providing more basic characteristic of patients would bring more interest from the readers.

According to Rewiever’s suggestion we provide a further table including basic characteristics of patients.

Minors:

Latin phrases like "in vitro" should go into italic font.

We modified into italic font the latin phrases.

Data Availability Statement should be filled up since there is a set of analyzed data presented.

We filled up Data Availability Statement section according to Reviewer’s suggestion.

The paper has multiple, redundant minor grammatical errors and very awkward phrases. Please proofread carefully. Clear and concise language is essential for effective scientific communication.

We carefully checked and revised the manuscript according to Reviewer’s suggestion.

Round 2

Reviewer 2 Report

Comments and Suggestions for Authors

The Authors addressed all of my concerns in a satisfactory way. The paper is good to go.